# Isolated *BAP1* Genomic Alteration in Malignant Pleural Mesothelioma Predicts Distinct Immunogenicity with Implications for Immunotherapeutic Response

**DOI:** 10.3390/cancers14225626

**Published:** 2022-11-16

**Authors:** Hatice Ulku Osmanbeyoglu, Drake Palmer, April Sagan, Eleonora Sementino, Michael J. Becich, Joseph R. Testa

**Affiliations:** 1Department of Biomedical Informatics, School of Medicine, University of Pittsburgh, Pittsburgh, PA 15206, USA; 2UPMC Hillman Cancer Center, Cancer Biology Program, School of Medicine, University of Pittsburgh, Pittsburgh, PA 15232, USA; 3Department of Bioengineering, School of Engineering, University of Pittsburgh, Pittsburgh, PA 15213, USA; 4Cancer Signaling and Microenvironment Program, Fox Chase Cancer Center, Philadelphia, PA 19111, USA

**Keywords:** tumor suppressor gene loss, BAP1, malignant pleural mesothelioma

## Abstract

**Simple Summary:**

Malignant Pleural Mesothelioma (MPM) is a highly aggressive, therapy-resistant cancer with a well-established inflammatory etiology and has no cure. Immune checkpoint inhibition (ICI) therapy has shifted treatment paradigms in many types of cancers, and recent clinical data have shown promise for improving MPM treatment. However, response to ICI therapy has been neither uniform nor predictable. The genomic landscape of MPM is primarily characterized by genomic alterations in tumor suppressor genes (TSGs) (~70%), particularly *BAP1*, *CDKN2A/B* and *NF2*. The impact of an isolated TSG genomic alteration versus multiple concurrent TSG alterations on clinical outcome, treatment response and MPM biology and the immune tumor microenvironment are unclear. Here, we showed the effect of TSG alteration combinations on clinical outcome, therapeutic response, and molecular pathways in MPM. For example, tumors with alterations in *BAP1* alone were (a) associated with a longer overall patient survival rate compared to tumors with *CDKN2A/B* and/or *NF2* alterations with or without *BAP1* and (b) comprised a distinct immunogenic subtype with altered transcription factor and pathway activity patterns.

**Abstract:**

Malignant pleural mesothelioma (MPM), an aggressive cancer of the mesothelial cells lining the pleural cavity, lacks effective treatments. Multiple somatic mutations and copy number losses in tumor suppressor genes (TSGs) *BAP1*, *CDKN2A/B*, and *NF2* are frequently associated with MPM. The impact of single versus multiple genomic alterations of TSG on MPM biology, the immune tumor microenvironment, clinical outcomes, and treatment responses are unknown. Tumors with genomic alterations in *BAP1* alone were associated with a longer overall patient survival rate compared to tumors with *CDKN2A/B* and/or *NF2* alterations with or without *BAP1* and formed a distinct immunogenic subtype with altered transcription factor and pathway activity patterns. *CDKN2A/B* genomic alterations consistently contributed to an adverse clinical outcome. Since the genomic alterations of only *BAP1* was associated with the PD-1 therapy response signature and higher *LAG3* and *VISTA* gene expression, it might be a candidate marker for immune checkpoint blockade therapy. Our results on the impact of TSG genotypes on MPM and the correlations between TSG alterations and molecular pathways provide a foundation for developing individualized MPM therapies.

## 1. Introduction

Malignant pleural mesothelioma (MPM) is a rare, devastating cancer of the lining of the lung and thoracic cavity with a 5-year survival rate of <10%. About 70% of MPM cases are associated with occupational or environmental exposure to asbestos fibers [1,2]. MPM is broadly divided into three histological subtypes with varying biological and clinical behaviors: epithelioid, sarcomatoid and biphasic, the latter being a combination of the former types [3]. The incidence of MPM has continued to increase in many parts of the world [4], and it is predicted to increase dramatically in certain developing countries such as India [5], where the unregulated use of asbestos has increased exponentially with few precautions taken. Moreover, carbon nanotubes, often used in manufacturing and for sportswear, induce inflammation as well as a cellular injury response similar to the in vitro response to asbestos fibers [6], which might lead to a future increase in MPM incidence [7].

The standard of care for MPM with pemetrexed platinum-based chemotherapy extends survival by only 2–3 months [8]. A combination of nivolumab and ipilimumab, the first drug regimen approved by the FDA for MPM since 2004, produced a four-month improvement in overall survival of MPM patients compared with those receiving standard of care cisplatin or carboplatin plus pemetrexed [9]. However, only a minority of MPM patients responded to immune checkpoint inhibitor (ICI) therapy. The results of the phase 3 CheckMate 743 study (NCT02899299) comparing first-line nivolumab and ipilimumab with chemotherapy revealed evidence of a greater ICI treatment effect among patients with non-epithelioid histologic subtypes compared to those with the epithelioid subtype [10]. Recent exploratory biomarker analyses with 3-year minimum follow-up from the CheckMate 743 study showed a significantly longer median overall survival (OS) among immunotherapy-treated patients with an inflammatory gene signature score (based on levels of CD8A, CD274/PD-L1, STAT1, and LAG3) regardless of histology [11]. To further understand ICI adoption in MPM with limited other therapeutic options, we need better understanding of this disease and effective molecular biomarkers for patient selection.

Loss-of-function mutations and copy number losses in tumor suppressor genes (TSGs) are common in MPM patients, whereas activating mutations of proto-oncogenes are rare [12,13,14,15]. Since it is difficult to develop therapies that target TSG alterations, little progress in a gene-targeted approach to MPM treatment has been made. The most common alterations of TSGs in the genomes of human MPM tumors are in *BAP1* (25–60% of cases), which encodes a deubiquitinating enzyme originally identified as a BRCA1 interacting protein; *CDKN2A/B* (40–45% of cases), which encodes the cell cycle inhibitors p16INK4A/p14ARF and p15INK4B, respectively; and *NF2* (20–50% of cases), which encodes the cytoskeletal scaffolding protein Merlin [12,13,14]. Although loss or inactivation of these TSGs occur in combination in ~35% of MPM cases, a recent evolutionary analysis by Zhang et al. [15] suggests that *BAP1* loss occurs early in the evolution of MPM, whereas *NF2* loss occurs later in disease progression. Among the TSGs commonly implicated in MPM, only *CDKN2A/B* has been associated with poor survival [12]. Further, OncoCast-MPM study show that TSG alterations in and of themselves are important factors in risk stratification of survival but in and of themselves may not be univariate predictors [16]. More recently, Hiltbrunner et al. divided pleural and peritoneal mesothelioma patients into four distinct subgroups according to alterations in *CDKN2A/B* and *BAP1* status using FoundationOne data [17].

The impact of single gene versus multiple TSG alterations on MPM biology, patient outcome, or treatment response is largely unknown. To address these gaps in our knowledge, we examined how single gene alteration of *BAP1*, *NF2*, or *CDKN2A* versus multiple TSG gene alterations affected clinical outcomes and response to therapy in MPM. We show herein that alteration of *BAP1* only is associated with a better outcome and a PD-1 therapy response signature and, thus, is a candidate biomarker for immune checkpoint blockade therapies. We also provide in silico evidence for altered transcription factors and pathways associated with each TSG/TSG combination genotype.

## 2. Results

### 2.1. Association of Patient Survival with TSG Genotypic Groups in MPM

To determine whether losses in each of these three genes or gene combinations could better stratify clinical outcomes, we analyzed MPM datasets from The Cancer Genome Atlas (TCGA) [12] (n = 86) and Memorial Sloan Kettering-Integrated Mutation Profiling of Actionable Cancer Targets (MSK-IMPACT, targeted screen) [18] (n = 61). The TCGA and MSK-IMPACT tumors were of mostly epithelioid histology (TCGA: epithelioid (66%), biphasic (26%), diffusive malignant (6%), sarcomatoid (2%); MSK-IMPACT: non specified (46%), epithelioid (44%), biphasic (7%), sarcomatoid (3%)) (Appendix A). All of the TCGA samples were surgical specimens from treatment naïve tumors, whereas MSK-IMPACT cohort consisted of a mix of resection samples and surgical specimens from tumors received pemetrexed-based cytotoxic therapy and/or checkpoint inhibitors.

We divided patients into eight groups based on deletions and/or mutations, “alteration” (denoted as ∆), based on frequent TSGs frequently involved in this disease: *BAP1* alteration alone (B∆; TCGA: n = 11, ~13%; MSK-IMPACT: n = 20, 33%), *NF2* alteration alone (N∆; TCGA: n = 7, ~8%; MSK-IMPACT: n = 2, ~3%), *CDKN2A/B* alteration alone (C∆; TCGA: n = 16, ~18%; MSK-IMPACT: n = 5, 8%), combined alterations in *CDKN2A/B* and *NF2* (N∆C∆; TCGA: n = 7, 8%; MSK-IMPACT: n = 3, 5%), alterations in *BAP1* and *CDKN2A/B* (B∆N∆; TCGA: n = 3, 3%; MSK-IMPACT: n = 6, ~10%), alterations in *BAP1* and *NF2* (B∆C∆; TCGA: n = 6, 7%; MSK-IMPACT: n = 3, ~5%), alterations in *BAP1*, *NF2*, and *CDKN2A/B* (B∆N∆C∆; TCGA: n = 11, ~13%; MSK-IMPACT: n = 2, ~3%), and alterations in genes other than these three driver TSGs (B+N+C+; TCGA: n = 26, 30%; MSK-IMPACT: n = 20, 33%) (Figure 1A,B). The main difference between the MSK-IMPACT and TCGA dataset appears to be the considerably higher incidence of alterations in *BAP1* alone seen in the MSK-IMPACT dataset (33%) versus the TCGA dataset (~13%).

We evaluated the association of overall survival (OS) with each TSG genotype by Kaplan–Meier analysis and univariate Cox proportional hazard models in the TCGA and MSK-IMPACT MPM datasets. Notably, tumors with an alteration of only *BAP1* (B∆) were significantly associated with longer survival compared with other tumors (32.28 months; HR 0.38 [95% CI 0.17–0.83]; *p* = 0.016 for TCGA and 21.14 months; HR 0.47 [95% CI 0.19–1.19]; *p* = 0·112 for MSK-IMPACT) (Figure 2A,B). B∆N∆ had lower hazard ratio in both TCGA and MSK-IMPACT studies but there were not enough samples to make a significant conclusion for this genotype. However, B∆N∆C∆ (8.32 months, HR: 2.62 [95% CI: 1.35–5.07], *p* = 0.004) and N∆C∆ (13.61 months, HR: 2.62 [95% CI: 1.32–6.78], *p* = 0.009) were significantly associated with poor survival in TCGA. When alterations in *CDKN2A/B*, *BAP1*, and *NF2* were modeled without interactions using a Cox proportional hazards regression model, tumors with *CDKN2A/B* alterations was associated with poor survival in both TCGA (HR: 3.23 [95% CI: 1.89–5.51], *p* = 2 × 10^−5^) and MSK-IMPACT (HR: 5.31 [95% CI: 1.70–16.57], *p* = 0.004) MPM datasets (Appendix A) as previously reported [12,16]. *BAP1* and *NF2* alterations had no significant effect on survival when modeled without interactions (Appendix A).

### 2.2. Association of Gene Signature Predictive of Response to Therapy with TSG Genotypic Groups in MPM

We performed gene set variation analysis (GSVA) [19] using published gene expression-based treatment-response signatures to determine differences in patient responses to standard pemetrexed chemotherapy and palbociclib based on TSG genotypes for the TCGA MPM dataset since parallel RNA-seq (RNA-sequencing) data was not available for theMSK-IMPACT cohort. We found an expression-based signature derived from non-small cell lung cancer that predicted resistance to pemetrexed [20] (Appendix A). We collected a signature of resistance to palbociclib derived from breast cancer [21] (Appendix A) to determine differences in sensitivity to cyclin-dependent kinase inhibitors among the eight TSG genotypic groups. Figure 3A shows GSVA scores for the two signatures across the MPM TSG tumors grouped by TSG genotypic types (see Materials and Methods). On average, MPM tumors with *CDKN2A/B* alteration, with or without *BAP1* or *NF2* alterations, were more resistant to pemetrexed and palbociclib, whereas tumors with only *BAP1* loss or with *BAP1* and *NF2* losses were more sensitive (a lower GSVA score).

We next compared immune checkpoint mRNA expression levels as a function of TSG genotypes and found that expression of *LAG3* (lymphocyte activation gene-3) and *VISTA* (V-domain Ig suppressor of T cell activation; also known *C10orf54*) was higher in B∆ tumors (Figure 3B), but mRNA levels for other immune checkpoint genes, including *PD-1, PD-L1*, *TIGIT*, and *CTLA4*, were not associated with TSG genotypes (Appendix A).

To evaluate the connection between TSG genotypes and resistance to PD-1 therapy, we also tested a gene expression signature predictive of benefit from immune checkpoint inhibitor (ICI) treatment [22] (Appendix A). Briefly, Jang et al. [22] generated the signature based on MPM patients treated with nivolumab (4 responders and 4 non-responders) and then they used it to subgroup TCGA samples into the anti-PD-1-responsive and anti-PD-1-resistant. Strikingly, we found that B∆ tumors were represented only in the anti-PD-1-responsive subgroup (*p* = 0.0032), whereas N∆C∆ tumors were observed only in the anti-PD-1 resistant subgroup (*p* = 0.0243) regardless of their histology (Figure 3C, Appendix A). We also applied CIBERSORTx to 86 TCGA tumors and inferred the infiltration of immune and tumor stromal cells using gene expression data. There was a lower proportion of B cells (*p*-value = 0.016, one-way ANOVA) and a higher proportion of natural killer (NK) (*p*-value = 0.031, one-way ANOVA) in B∆ tumors compared to other TSG genotype groups (Appendix A). However, we did not observe differences for other cell types including CD8+ T cells. In summary, tumors with only *BAP1* alteration (B∆) without other frequent TSG alterations formed a distinct MPM subtype that was associated with significantly longer overall patient survival, a better therapeutic response signature (e.g., anti-PD-1 and pemetrexed), and higher expression of *LAG3* and *VISTA*.

### 2.3. Transcription Factors and Pathways Associated with TSG Genotypic Groups in MPM

To determine whether the eight MPM TSG genotypes impacted similar or distinct molecular pathways, we performed sample-specific transcription factor activity analysis via the Integrated System for Motif Activity Response Analysis (ISMARA) [23] and sample-specific pathway enrichment analysis via GSVA [19] using gene expression data based on TCGA RNA-seq data. Figure 4A summarizes mean transcription factor (TFs) activities (false discovery rate [FDR] < 0.05) and mean pathway scores (FDR < 0.05) significantly associated with TSG genotypes. Tumors with *CDKN2A* alteration including B∆N∆, B∆N∆C∆ and N∆C∆ were associated with increased activity of E2F Targets (related to cell cycle) (Figure 4A,B). Further, B∆N∆C∆, B∆N∆, C∆, and N∆ tumors were associated with increased activity of epithelial-mesenchymal transition (EMT) (Figure 4A,B). Whereas B∆ tumors were associated with increased activity of interferon regulatory factors (IRFs) that have a role in immunity and decreased activity of *BCL6B* (also known as B-cell CLL/lymphoma 6 member B), which is a transcriptional repressor that interacts with the Notch, STAT, p53 and PI3K/AKT signaling pathways, all of which may be involved in inflammatory response regulation in cancer cells [24,25,26] (Appendix A). Consistent with the TF activity patterns, B∆ tumors were associated with increased interferon-alpha and interferon-beta signaling activity. Genes that showed increased expression that correlated with IRF TF activities and pathway score included interferon response genes, such as *IFIT3*, *OAS2*, *OAS3*, *IFIH1*, *STAT1*, *DDX60*, and *DDX58*; genes regulating the CGAS-induced type I interferon signaling pathway, such as TRIM14; and transporters associated with antigen processing such as *TAP1* and *TAP2* (Figure 4C). In summary, we identified a group of MPM patients with loss-of-function mutations and/or copy number loss only in B∆ tumors, which define a distinct molecular subtype associated with a high interferon response.

## 3. Discussion

Precision oncology, which has been successful in treating a variety of cancers, is still in its infancy in MPM. While there are ongoing clinical trials for MPM, there are no defined molecular markers (e.g., genomic markers) that can be used to predict the efficacy of treatments. Genomic alterations of *BAP1*, *NF2*, and *CDKN2A/B* TSGs are thought to play a critical role in MPM pathogenesis. In this study, we provide a comprehensive analysis of data on MPM tumors with genomic alteration in one or more of these specific TSGs, which are thought to be critical drivers of MPM pathogenesis.

Tumors with alterations in only *BAP1* showed a distinct pattern of expression of inflammatory tumor microenvironment genes, including activation of interferon signaling and IRF TFs and high *LAG3* and *VISTA* expression. Interferon production is a defense response that recruits and activates immune cells and has been studied extensively in cancer. Activation of the interferon response in cancer cells [27], possibly by genomic instability through the cGAS–STING pathway [28], may affect immune cells in the tumor microenvironment and the tumor response to immunotherapies. In tumors, type I interferons are secreted by cancer cells and dendritic cells (DCs) in response to DNA fragments that activate the cGAS/STING pathway and result in T cell priming and antitumor activity. Thus, isolated *BAP1* alteration may serve as a predictive and prognostic candidate biomarker for MPM to improve disease stratification and therapy. Interestingly, *BAP1* alterations have recently been shown to be correlated with perturbed immune signaling in malignant peritoneal mesothelioma [29]. In addition, in vitro and in vivo studies have shown type I interferon (IFN-I) activation in mesothelioma cells [30,31]. Further, Hmeljak et al. [12] reported an IFN-I signature in pleural mesothelioma tumors with an inactivated *BAP1* gene. Further, Yang et al. showed a negative correlation between *BAP1* expression and a constitutively activated IFN-I response [32]. Regarding possible future validation experiments to confirm the tumor-promoting roles of IRF TFs and interferon signaling in B∆ MPM, expression levels of IRF TFs can be manipulated in cultured cells through overexpression or silencing, and the effects of IRF TFs on cancer cell proliferation, survival, motility, and invasion potential can then be evaluated. Ultimately, in vivo validation of the roles of IRF TFs expression on MPM progression can be studied using mouse models of mesothelioma crossed with mice harboring knockout of specific IRF TF genes [33].

Harnessing the combination of different immunotherapy approaches to improve outcomes of patients with MPM is an area of clinical interest [34]. We observed that expression of immune checkpoint genes *LAG3* and *VISTA* were higher in B∆ tumors. VISTA is a member of the B7 family of B7-CD28 family of ligands and receptors that is expressed primarily on myeloid cells and T-lymphocytes [35,36]. When overexpressed, VISTA suppresses early T-cell activation and proliferation and reduces cytokine production. Hmeljak et al. [12] reported strong expression of VISTA in benign mesothelium by immunohistochemistry (IHC) and increased mRNA expression of VISTA in epithelioid MPM compared to other tumor types. Muller et al. [37] also confirmed VISTA expression by IHC in malignant pleural mesotheliomas in both tumor and infiltrating inflammatory cells. LAG3 is a coinhibitory receptor expressed on activated T cells and has now become part of the repertoire of combinatorial immunotherapeutics available for the treatment of metastatic melanoma [38]. Recently, Marcq et al. [39] reported monotherapy with PD-L1 and its combination with LAG-3 blockade, resulted in delayed tumor growth and significant survival benefit in malignant mesothelioma mouse models. Our study also provides a candidate biomarker for preclinical studies of PD-1 combination therapy with a VISTA or LAG3 inhibitor in B∆ MPM tumors.

A major limitation of our study is our small cohort size. Further, there are several differences in percentages for the different TSG status between the TCGA and the MSK-IMPACT datasets. One reason might be due to the fact that the TCGA data was based on whole exon sequencing whereas MSK-IMPACT was based on a targeted screen, which may have identified a higher percentage of whole exon deletions of *BAP1*. Notably, in early reports, performed with Sanger sequencing, revealed point mutations in 20–25% of sporadic MPMs [40,41]. Subsequent deletion mapping and multiplex ligation-dependent probe amplification studies have identified alterations of BAP1 in 50–60% of MPMs, with the increase due to inactivating deletions of entire *BAP1* exons [42,43]. Despite the limitations, our results indicate that isolated *BAP1* could be candidate biomarker for selection of patient to immune checkpoint blockade therapies.

## 4. Materials and Methods

### 4.1. Data and Preprocessing

TCGA data were downloaded through the Broad Institute TCGA GDAC firehose tool. RNA-seq data were available for 86 samples. Genetic alteration data (copy number alteration and mutation) for *CDKN2A/B*, *NF2*, and *BAP1* were retrieved from cBioPortal and clinical data from an online portal for data from the TCGA project and MSK-IMPACT [18].

### 4.2. Survival Analyses

Overall survival (OS) was defined as the time from diagnosis to death resulting from any cause. Survival curves were estimated using the Kaplan–Meier method. Hazard ratios (HR) were estimated using a Cox proportional hazards regression model. For each of the eight TSG genotypes, a separate univariate regression model was created with one binary variable indicating if a tumor had the exact genotype. To model the effect of genotypes on survival without interactions, a Cox proportional hazards regression model was created with three binary variables, one for each: *CDKN2A/B*, *BAP1*, and *NF2*. *p*-values were obtained for each coefficient in the Cox regression models with a Wald-test, with the null hypothesis that the HR is 1. Both the Cox regression and Kaplan–Meier analyses were done with the survival R package [44].

### 4.3. Motif Activity Analysis

To analyze activities of transcription factor binding motifs (TFBM) from TCGA MPM RNA-seq data, we used ISMARA [23].

### 4.4. Gene Set Enrichment Analysis

GSVA and single-sample gene set enrichment analysis (ssGSEA) were performed using the GSVA R package (version 1.40.1) [19] on the pemetrexed and palbociclib response signature (Appendix A) and pathway enrichment analysis. For pathway enrichment analysis, we obtained pathway annotations from the Molecular Signatures Database (MsigDB) [45], a collection of hallmarks of cancer and REACTOME pathways (c2.all.v7.1.symbols.gmt). The log-transformed TMM (trimmed mean of M values) normalized TPM (transcripts per million) counts based on TCGA MPM RNA-seq data were used as input to the GSVA package.

### 4.5. Analysis of the Tumor Immune Microenvironment (TIME)

The proportion of different immune, tumor and stromal cells in the tumor microenvironment was estimated from RNA-seq data using CIBERSORTx [46]. The CIBESORTx algorithm was run with default settings, excluding quantile normalization, for 100 permutations with our signature matrix based MPM CITE-seq data [47] from one tumor sample (GSE172155) to estimate the abundance of immune, stromal and malignant cells types (Appendix A). Then, we evaluated the association of cell types with each TSG genotype using one-way ANOVA.

### 4.6. Statistical Analysis and Visualization

All statistical tests in the exploratory analysis were performed using R version 4.1.1 and associated packages. The statistical analyses for differences in mRNA expression, GSVA score, and TF activity across TSG genotype groups were performed using one-way ANOVA with an FDR cut off of 0.05. Further, we used one-way ANOVA and post hoc Tukey’s HSD (honestly significant difference), with an adjusted *p*-value cut off of 0.1 for pairwise comparison of genotypes for visualization with boxplots. One or more letters were assigned to each TSG genotype group using the multcompView package in R (version: 0.1–8). Assignments were made such that any two samples that had a statistically significant difference did not share any letters.

Graphs were generated using RColor-Brewer (version: 1.1 2), ggplot2 (version: 3.3.3) ComplexHeatmap (version: 2.4.3), ggrepel (version: 0.9.1), and circlize (version: 0.4.13) packages. For general data analysis and manipulation, dplyr (version: 1.0.7), matrixStats (version: 0.59.0) and data.table (version: 1.14.0) were used.

## Figures and Tables

**Figure 1 cancers-14-05626-f001:**
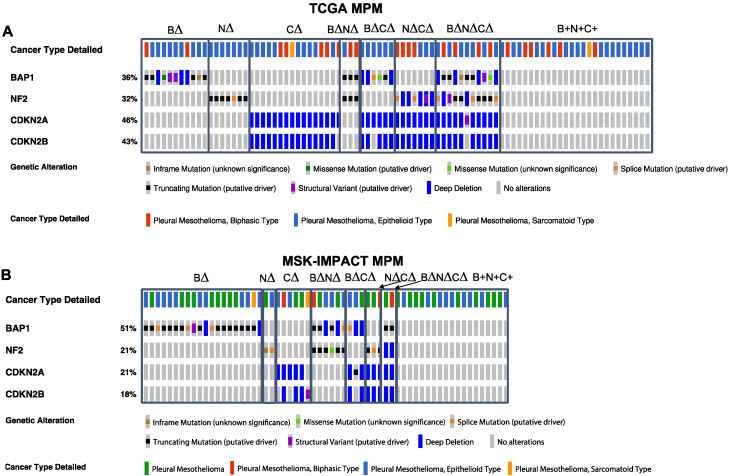
*BAP1*, *NF2*, and *CDKN2A/B* tumor suppressor gene (TSG) genomic alterations in human malignant pleural mesothelioma (MPM) cohorts: (**A**) TCGA and (**B**) MSK-IMPACT. *BAP1* (B), *NF2* (N), and *CDKN2A* (C). Deletions and/or mutations collectively denoted as ∆.

**Figure 2 cancers-14-05626-f002:**
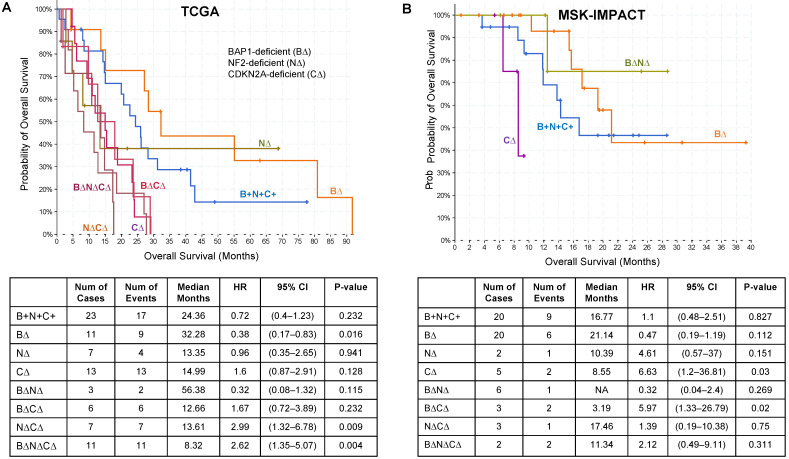
The association of TSG genotypes with overall survival in MPM. Kaplan–Meier plots of clinical outcomes based on *BAP1* (B∆), *NF2* (N∆), and *CDKN2A* (C∆) TSG genotype combinations. B+N+C+ refers to tumors that have alterations of genes other than *BAP1*, *NF2*, or *CDKN2A/B*. TCGA (**A**) and MCK-IMPACT (**B**) MPM datasets. We filtered groups with less than five samples from Kaplan–Meier plots.

**Figure 3 cancers-14-05626-f003:**
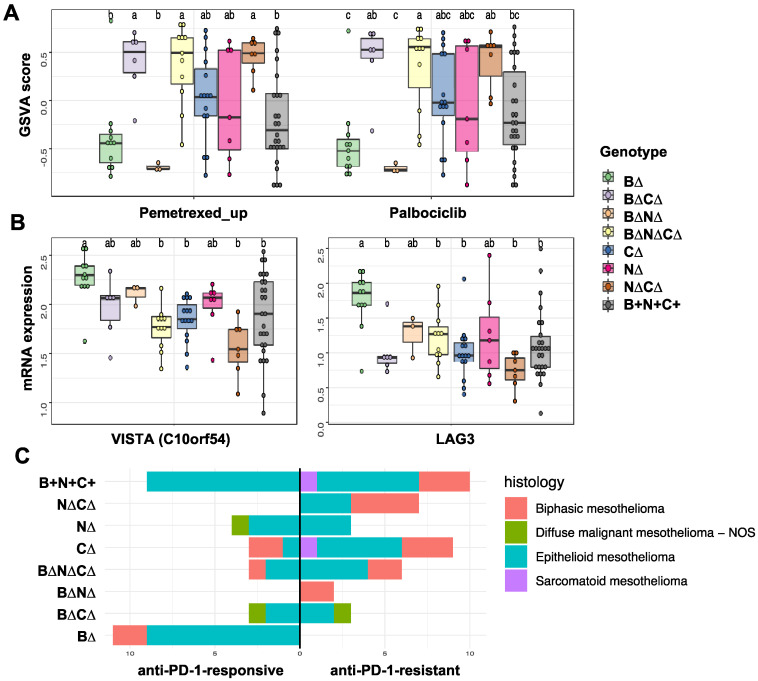
The association of TSG genotypes with drug response signatures in MPM. (**A**) Enrichment for signatures of resistance to chemotherapy and targeted therapy in patients with MPM (pemetrexed, left panel; palbociclib, right panel). Positive versus negative GSVA scores (y-axis) indicate upregulation or downregulation of the signature in each tumor. Samples are grouped by TSG genotype. (**B**) Comparison of immune checkpoint mRNA expression levels as a function of TSG genotypes in 86 MPM samples from the TCGA cohort. We use the one-way analysis of variance (ANOVA) analysis (FDR < 0.05) for statistical analysis. Samples with different letters exhibited statistically significant mRNA expression or GSVA score differences (ANOVA, Tukey’s HSD, adjusted *p*-value < 0.1). (**C**) The anti-PD-1-resistant mRNA signature was used to predict the subgroups. TCGA MPM tumors predicted to be anti-PD-1-sensitive tumors were enriched in samples with *BAP1* genomic alteration only. The bar plot shows number of samples in each group by histological type. The data with different little letters show significant difference.

**Figure 4 cancers-14-05626-f004:**
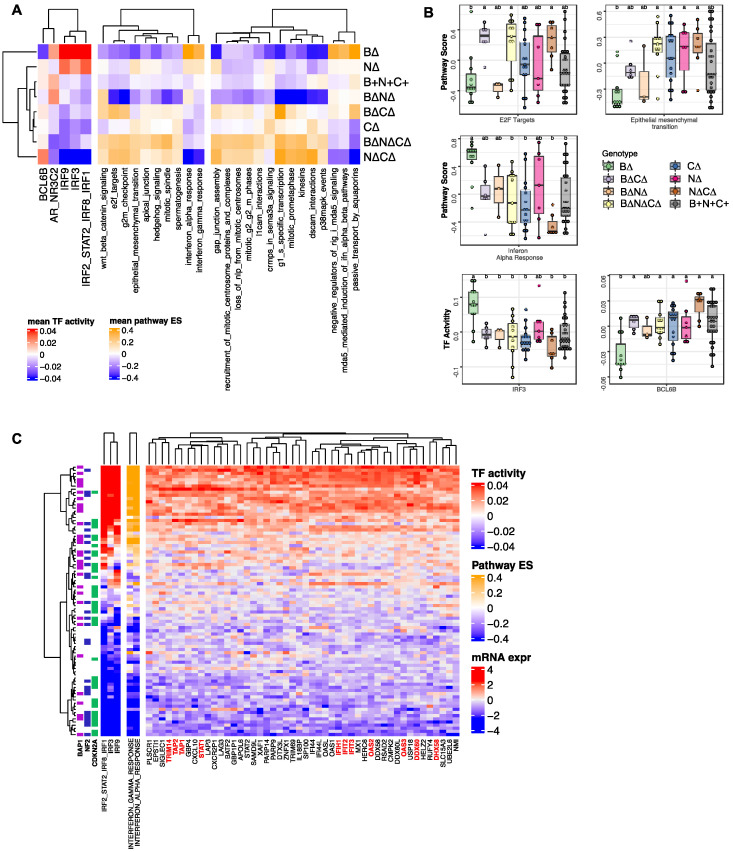
The association of TSG genotypes with TF activity and pathway patterns. (**A**) Heatmap showing mean sample-specific TF activity and pathway enrichment scores that are significantly associated with TSG genotype groups based on the one-way ANOVA analysis (FDR < 0.05). (**B**) Boxplots indicate the distribution of inferred IRF3, and BCL6 TF activities and interferon-alpha response, E2F targets and epithelial to mesenchymal transition enrichment score (ES) based on gene expression profiles across tumor TSG genotypes. Samples with different letters exhibit statistically significant TF activity or pathway enrichment GSVA score (one-way ANOVA, Tukey’s HSD, *p* < 0.1). (**C**) The top heat map shows tumors clustered by the inferred IRF family TF activities. The middle panel shows the pathway ES for interferon pathways for each tumor based on clustering by TF activities. The bottom panel shows the mRNA expression profile for each tumor for genes highly correlated with IRF TF activity (absolute value of Pearson correlation > 0.75). The data with different little letters show significant difference.

## Data Availability

Publicly available datasets were analyzed in this study. This data can be found here: (1) RNA-seq gene expression data from TCGA’s Firehose data run (https://confluence.broadinstitute.org/display/GDAC/Dashboard-Stddata, accessed on 28 September 2022); (2) Genetic alteration data (copy number alteration and mutation) and clinical data were retrieved from cBioPortal (http://www.cbioportal.org) [48] for both the TCGA project and MSK-IMPACT [18].

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
