# Peer review of "Isolated BAP1 Genomic Alteration in Malignant Pleural Mesothelioma Predicts Distinct Immunogenicity with Implications for Immunotherapeutic Response"

_cancers, 2022, doi:10.3390/cancers14225626_

Round 1
Reviewer 1 Report
The authors present a concise research paper in which they assess the impact of single vs. multiple alterations to three canonical TSG (BAP1, CDKN2A and NF2) on pleural mesothelioma biology; superficially identifying an association in patients with alterations to only BAP1 and improved outcome as well as the potential for single BAP1 alterations as a signature for likely response to PD-1 immune checkpoint blockade.
The paper is well written, appropriately referenced and will be of interest to the mesothelioma and wider cancer research communities. The authors performed complex genomic analyses of well‑characterised mesothelioma datasets to stratify patients into eight subsets based on single of multiple TSG alterations across two distinct datasets. Data analysis is well performed, and conclusions supported by the data presented. Limitations are clearly stated.
In general, the article is of publication standard, subject to correction of details below.
Suggested corrections / clarification.
References: Line 50 ref 10. Would be original Baas Lancet Checkmate743 study be more appropriate here?
Formatting: First paragraph of results chapter (Lines 84 -91). Percentage symbol precedes the number (i.e., %66), while the remainder of the manuscript is in the standard format. Please rectify for consistency.
Suggest changing nomenclature for deletion from using minus symbol to using ‘delta’ to denote deletion. (i.e. C- = ∆C).
Figure 2: Kaplan Myer survival plots. Not statistical information is provided on either the plot or the associated data table. Given that the authors are state that tumours with only BAP1 deletion are significantly associated with longer survival compared to those with CDKN2A and/or NF2, this should be shown. Also, there is no comment on NF2 only deletion, which seems to have a similar outcome to BAP1 only deletions. Further comment/discussion should be provided.
Figure 4: Discussion of figure 4 in text relates to data that is not shown in the figure, and likewise, data in the figure is not mentioned. Please clarify.
Reviewer 2 Report
The manuscript has innovative data regarding genomic alterations associated with immune checkpoint (IC) therapy response in Malignant pleural mesothelioma, an aggressive cancer lacking effective treatments. This manuscript fits perfectly with the research topic of the journal. However, I have a minor comment that I think should be addressed in this kind of research where data is publicly available, and no de novo data is generated.
From an immunological point of view, although data is from bulk and no scRNAseq data is available for this dataset, I would suggest annalizing data with either ABBAS, CIBERSORT, MIXTURE, or other similar tools to try to describe better the tumor microenvironment and how IC blockade response correlates with tumor mutations. If not, the link is not completely understood.
Reviewer 3 Report
The article by Osmanbeyoglu et al., entitled “Isolated BAP1 genomic alteration in malignant pleural mesothelioma predicts distinct immunogenicity with implications for immunotherapeutic response”, elucidates how the impact of single gene alteration of BAP1, NF2, or CDKN2A versus multiple TSG alterations affected clinical outcomes and response to therapy in MPM. In particular, the authors showed that alteration of BAP1 only is associated with a better outcome and a PD-1 therapy response signature and, thus, could represent a candidate biomarker for immune check-point blockade therapies. Moreover, they provide evidence for altered transcription factors and pathways associated with each TSG/TSG combination genotype.
The authors are aware of some limitations to their work: the small cohort size and the differences in percentages for the different TSG status between the TCGA and the MSK-IMPACT datasets.
Overall the in silico analysis is well written and expanded, but there are some points to review:
- In my point of view, another limitation of the present work is the absence of a deeper discussion on the results herewith obtained, making the manuscript a general in silico study to be further validated. The authors identified new interesting associations among TSG genotypic groups and specific transcription factors and/or pathways, for example, but since none in vitro/in vivo validation has been presented, I suggest that the authors at least deepen the discussion for the identified associations in the attempt to speculate/hypothesize about the underlying mechanisms and suggesting the need for further experiments.
- Discussion section. line 218: “Corroborating our findings…”
The authors use literature’s evidences to corroborate their in silico results rather than corroborate literature’s data with their new findings. I suggest to the authors to rephrase the entire sentence.
- I would like to suggest to the authors to pay more attention to figures preparation and results presentation:
Figure 1. Please, add in figure 1B the subdivision into the 8 groups of patients as in 1A.
Figure 2. Please, add in both tables the p-value from log-rank test. Moreover, we observe something wrong with 95% confidence interval limits: why the major one is always NA? Please, add the explanation for this event in the main text and the detailed analysis methods in M&M section. Finally, the authors declare in the figure legend that they filtered groups with less than five samples for survival analysis, but in the figure 2B is absent the group B-N- with 6 patients. Why? Please, correct the figure.
Figure 3.
3A. line 155-156: Positive versus negative GSVA scores (y-axis) indicate upregulation or downregulation of the signature in each patient, respectively. Do you mean: in each patient group? Please, correct the sentence. Which dataset was used for the analysis? TCGA o MSK-IMPACT? Or both? Please, specify it also in the main text.
3A-B. Line 159: adjusted P-value < 0.1. The authors should consider this minimum stringency for the discussion of results.
3C. To evaluate the connection between TSG genotypes and resistance to PD-1 therapy, the authors used TCGA MPM tumors dataset. It would be interesting to reproduce the same analysis also using the MSK-IMPACT datasets. Did the authors already consider this option? It would be interesting to discuss.
Figure 4.
4A. Line 168-169. The figure summarizes a set of transcription factors (TFs) (false discovery rate [FDR] < 0.1) and pathways (FDR < 0.05) associated with each TSG group. It would be advisable to standardize the FDR values (preferably to < 0.05).
4B. Genotype group: please correct N- instead of N0.
- Results section: “Association of gene signature predictive of response to therapy with TSG genotypic groups in MPM”
I suggest to the authors to add a supplementary file with the molecular signatures from literature used in this work to compare the two datasets of interest.
Reviewer 4 Report
Osmanbeyoglu et al. presented an interesting and complex work, even if a little bit not so innovative, concerning the new emerging strategies in the molecular and genetic approach against malignant pleural mesothelioma (MPM), particularly analysing the impact of tumor suppressor genes BAP1, CDKN2A/B, and NF2 on MPM and the new possible therapies in association with the genomic heterogeneity of this rare cancer.
The authors have focused their results on the BAP1 gene, by demonstrating a correlated and interesting new approach against MPM, so this reviewer requires no essential changes to the work, apart minor revisions concerning some syntax or typing errors (i.e.: page 1, lines 36; etc.) in the document, which the authors will need to correct.
Round 2
Reviewer 2 Report
The manuscript has been substantially improved with all reviewers suggestions and all my concerns have been addressed!
Reviewer 3 Report
The manuscript has been properly finalized according to the required revision(major). I endorse its publication in the present form.